# Guideline-level monitoring, biomarker levels and pharmacological treatment in migrants and native Danes with type 2 diabetes: Population-wide analyses

**Anders Aasted Isaksen**[1,2]*, **Annelli Sandbæk**[1,2], **Mette Vinther Skriver**[1], **Gregers Stig Andersen**[4], **Lasse Bjerg**[2,3]

**1** Department of Public Health, Aarhus University, Aarhus C, Denmark, **2** Steno Diabetes Center, Aarhus, Denmark, **3** Department of Paediatrics, Viborg Regional Hospital, Viborg, Denmark, **4** Steno Diabetes Center Copenhagen, Herlev, Denmark

* aai@ph.au.dk

**Data Availability Statement:** The data is owned by and available through Statistics Denmark and the Danish Healthdata Authority, although fees apply

## Abstract

The prevalence of type 2 diabetes (T2D) is higher in migrants compared to native populations in many countries, but the evidence on disparities in T2D care in migrants is inconsistent. Therefore, this study aimed to examine this in Denmark. In a cross-sectional, register-based study on 254,097 individuals with T2D, 11 indicators of guideline-level care were analysed: a) monitoring: hemoglobin-A1c (HbA1c), low-density lipoprotein cholesterol (LDL-C), screening for diabetic nephropathy, retinopathy, and foot disease, b) biomarker control: HbA1c and LDL-C levels, and c) pharmacological treatment: glucose-lowering drugs (GLD), lipid-lowering drugs, angiotensin-converting enzyme-inhibitors/angiotensin receptor blockers, and antiplatelet therapy. Migrants were grouped by countries of origin: Middle East, Europe, Turkey, Former Yugoslavia, Pakistan, Sri Lanka, Somalia, Vietnam. In all migrant groups except the Europe-group, T2D was more prevalent than in native Danes (crude relative risk (RR) from 0.62 [0.61–0.64] (Europe) to 3.98 [3.82–4.14] (Sri Lanka)). In eight indicators, non-fulfillment was common (>25% among native Danes). Apart from monitoring in the Sri Lanka-group, migrants were at similar or higher risk of non-fulfillment than native Danes across all indicators of monitoring and biomarker control (RR from 0.64 [0.51–0.80] (HbA1c monitoring, Sri Lanka) to 1.78 [1.67–1.90] (LDL-C control, Somalia)), while no overall pattern was observed for pharmacological treatment (RR from 0.61 [0.46–0.80] (GLD, Sri Lanka) to 1.67 [1.34–2.09] (GLD, Somalia)). Care was poorest in migrants from Somalia, who had increased risk in all eleven indicators, and the highest risk in nine. Adjusted risks were elevated in some migrant groups, particularly in indicators of biomarker control (fully-adjusted RR from 0.84 [0.75–0.94] (LDL-C levels, Vietnam) to 1.44 [1.35–1.54] (LDL-C levels, Somalia)). In most migrant groups, T2D was more prevalent, and monitoring and biomarker control was inferior compared to native Danes. Migrants from Somalia received the poorest care overall, and had exceedingly high lipid levels.

and researchers must be affiliated with an approved research institute in Denmark. Information on requesting data can be found at https://www.dst.dk/en/TilSalg/Forskningsservice.

**Funding:** Financial support provided to AAI by Steno Diabetes Center Aarhus, which is partially funded by an unrestricted donation from the Novo Nordisk Foundation, and by Public Health in Central Denmark Region - a collaboration between municipalities and the region (grant no. A2436).

**Competing interests:** AS and LB are employees of Steno Diabetes Center Aarhus, which is partially funded by an unrestricted donation from the Novo Nordisk Foundation. This does not alter our adherence to PLOS ONE policies on sharing data and materials.

## Introduction

High prevalences of type 2 diabetes (T2D) have been observed in migrant groups in Denmark [1] and other European countries [2]. T2D care quality may differ between migrants and the native populations, and previous studies on T2D care quality in Denmark have shown worse glycaemic control in migrants compared to native Danes despite similar levels of diabetes monitoring [3, 4]. In other European countries, inferior glycemic control in migrants and ethnic minority groups has been documented in several studies [5–8], but studies of other aspects of type 2 diabetes care have shown heterogeneous findings: In studies from Sweden and Norway, migrants with T2D received similar or more monitoring and were more likely to receive glucose-lowering medication than the native populations [5, 7]. Contrarily, a study from Italy found that migrants with drug-treated T2D received fewer medications and referrals for consultations than native Italians with T2D [9]. In England, Blacks and Asians were less likely to receive monitoring for diabetic retinopathy than Whites, and Blacks received less testing for hemoglobin A1c (HbA1c) and were less likely to be prescribed sodium-glucose cotransporter-2 inhibitors and glucagon-like peptide-1 agonists [8]. Also, South Asians and African/Caribbean minorities were less likely to be prescribed statins than ethnic Europeans in England [10].

To the best of our knowledge, no study has investigated all of these aspects of T2D care in a population-wide, national setting.

For all citizens and migrants with residence permits, healthcare is provided within the Danish public healthcare system free of charge, although T2D patients incur some out-of-pocket copayments for podiatrist care and prescription drug purchases at pharmacies [11]. Patients with T2D are primarily treated by the general practitioners, which native Danes and migrants with residence permits have free access to. The Danish College of General Practitioners publish national clinical guidelines for T2D (CG) in cooperation with the Danish Endocrine Society [12]. While CG advise that monitoring intervals, biomarker goals and treatment intensity is adapted to fit the individual patient, they contain several specific recommendations that may be used as indicators of care quality. These recommendations include yearly monitoring of HbA1c and low-density lipoprotein cholesterol (LDL-C) levels, as well as urine albumin-to-creatinine ratio (UACR) screening for diabetic nephropathy. CG also recommend screening for diabetic retinopathy by an ophthalmologist every second year and screening for diabetic foot disease by a podiatrist every year. At the initial diagnosis of T2D, a baseline-screening of all five types of monitoring is recommended. For most individuals without complications, CG recommend HbA1c below 53 mmol/mol and LDL-C below 2.6 mmol/L. Pharmacological treatment with glucose-lowering drugs (GLD) is recommended in individuals with HbA1c $\geq$ 48 mmol/mol, and lipid-lowering drugs (LLD) in individuals with prevalent cardiovascular complications, diabetic nephropathy, or LDL-C above 2.5 mmol/L. Angiotensin-converting enzyme-inhibitors or angiotensin receptor blockers (ACEI/ARB) are recommended in individuals with prevalent macrovascular complications or diabetic kidney disease (DKD), as is antiplatelet therapy (APT). No migrant-specific recommendations are provided for these areas of care, but CG advise taking ethnicity into account as a risk factor when screening for T2D.

To map the quality of T2D care in these vulnerable groups and highlight areas with potential for improvement, the aim of this study was to assess prevalence of T2D and disparities between migrant groups and native Danes across eleven guideline-recommendations of T2D care covering monitoring, biomarker levels and pharmacological treatment.

## Materials and methods

### Definition of study populations

The background population included all adults residing in Denmark on the index date (1 January 2018). To address limitations and ease clinical interpretation of findings, T2D prevalence was analysed after the following exclusions: i) individuals younger than 25 years or older than 99 years, where register-based diabetes-classification may be inaccurate due to different clinical management, ii) second-generation descendants of immigrants, who were outside the scope of this study, iii) immigrants residing in Denmark for less than 3 years on the index date, in whom data may be incomplete, and iv) migrants from countries not in the selected origin categories. Fig 1 shows the flow of the study population.

Analyses of T2D care quality were performed in a subset of this population containing only those with T2D onset at least 6 months prior to the index date in order to allow time for baseline-screening, initiation of pharmacological treatment and subsequent biomarker control to occur in the newly-diagnosed. Within this subset, monitoring was evaluated among all individuals, and biomarker levels evaluated in those with at least one biomarker measurement prior to the index date. Pharmacological treatment was evaluated in a subset of individuals alive and resident in Denmark in the year following the index date, where CG recommend treatment according to complication status or biomarker levels on the index date: i) GLD: HbA1c ≥48 mmol/mol at the most recent measurement. ii) LLD: prevalent macrovascular complications or DKD, or age above 40 with LDL cholesterol above 2.5 mmol/L. iii) ACEI/ARB and APT: prevalent macrovascular complications or DKD.

### Data sources

Data for this study was gathered from several sources and linked on an individual level using the unique personal identifier given to all Danish residents at immigration or birth. Civil registration, all healthcare contacts and drug prescriptions in Denmark are recorded with this identifier, thus enabling complete data linkage and coverage of the entire population.

**Healthcare data.** Information on all hospital admissions and outpatient contacts from 1994 through 2018 was obtained from the Danish National Patient Register (NPR) [13].

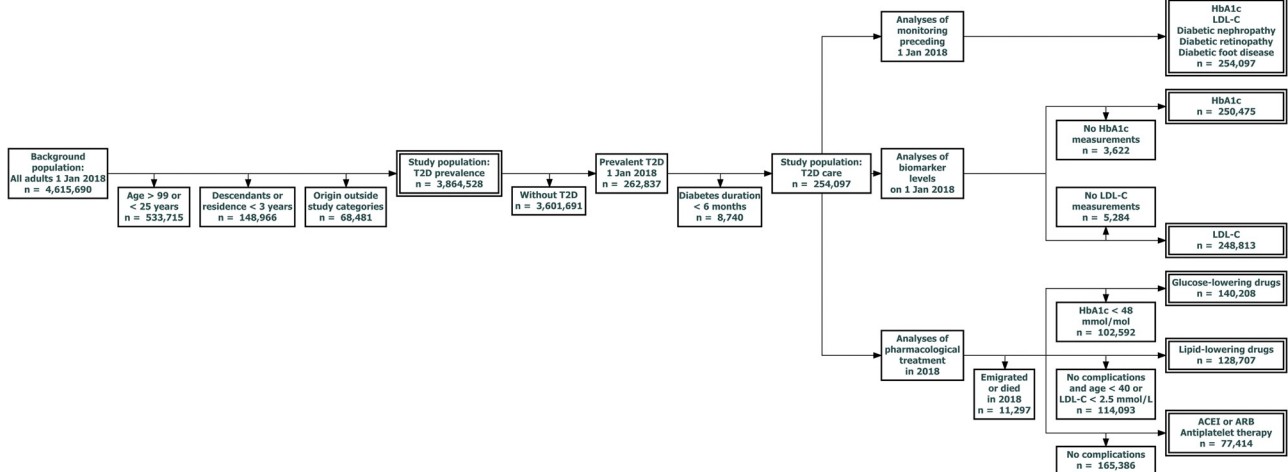

**Fig 1. Flow of study populations.** Double-bordered boxes indicate populations used in analyses. T2D, type 2 diabetes; HbA1c, hemoglobin-A1c; LDL-cholesterol, low-density lipoprotein cholesterol; ACEI, angiotensin-converting enzyme-inhibitors; ARBs, angiotensin receptor blockers.

Information on all diabetes-specific services performed at general practitioners, podiatrists and ophthalmologists in the primary sector from 1990 through 2018 was obtained from the Danish National Health Service Register (NHSR) [14]. Information on all medications use from 1997 through 2018 was obtained from the Danish National Prescription Registry (NPrR) [15], classified according to the Anatomical Therapeutic Chemical (ATC) Classification. Information on all routine clinical biomarker samples was obtained from the Register of Laboratory Results for Research (RLRR) and the Clinical Laboratory Information System (CLIS) of Central Denmark Region (see S1 File for specific sample codes). In combination, the biomarker data covered all residents in Denmark from 2015 through 2018 and four of five Danish administrative regions were covered from 2011 through 2018.

**Demographic and socioeconomic data.** Information on age, sex, equivalised disposable household income [16], employment status, region of residence, migrations and country of origin was obtained from the Danish Civil Registration System (CRS) [17].

## Definition of study outcomes

For the analysis of T2D prevalence, T2D prevalent on the index date was used as the outcome.

T2D was defined using a previously validated classifier, the *Open-Source Diabetes Classifier* [18], based on the following inclusion events: i) HbA1c measurements of $\geq$48 mmol/mol (6.5%), ii) hospital diagnoses of diabetes, iii) diabetes-specific podiatrist services, iv) purchases of GLD. Diabetes type was classified as either type 1 or T2D based on patterns of insulin purchases and type-specific hospital diagnoses of diabetes.

In a series of cross-sectional analyses, T2D care quality was evaluated in the scope of failure to meet the recommendations of CG [12] in terms of monitoring (allowing for three months of extra leeway), biomarker levels, and pharmacological treatment:

Monitoring of HbA1c, LDL-C, and UACR-screening were identified by records of biomarker samples in RLRR/CLIS between 1 October 2016 and 31 December 2017. Monitoring of HbA1c with point-of-care-testing in the primary care sector was identified by records in NHSR. Similarly, screening for diabetic foot disease was identified by records of diabetes-specific podiatrist services received between 1 October 2016 and 31 December 2017. Diabetic retinopathy screening was identified by records of retinal photoscreening for diabetic retinopathy at hospital departments (in NPR) or diabetes-specific examinations at ophthalmologist practice between 1 October 2015 and 31 December 2017 (to match the two-year window recommended for retinopathy screening in Danish guidelines).

The most recent measurement of HbA1c and LDL cholesterol in RLRR/CLIS before the index date was used to evaluate biomarker levels. The outcome measure for HbA1c was a level $\geq$53 mmol/mol (7.0%). For LDL cholesterol, the outcome measure was a level $\geq$2.6 mmol/L.

Four types of pharmacological treatment were assessed in the year following the index date, and treatment was defined as at least one purchase of that type of drug based on anatomical therapeutic chemical (ATC) codes (including subcodes): i) GLD: ATC codes A10. ii) LLD: ATC codes C10. iii) ACEI/ARB: ATC codes C09A-C09D. iv) APT: ATC codes B01AC.

## Definition of exposures and explanatory variables

Immigrant status and country of origin was defined according to the definition of migrant origin by Statistics Denmark based on country of birth and citizenship of parents [19]. Based on the national origins with most T2D cases and the United Nations M49 geoscheme, the population was grouped into 9 origin categories: Native Danish, Europe, Middle East (M49: Northern Africa and Western Asia, plus Afghanistan and Iran), and the six remaining individual

countries with most T2D cases: Turkey, Former Yugoslavia, Pakistan, Sri Lanka, Somalia, and Vietnam. Individuals from these countries were not included in the Middle East or Europe groups. Age and diabetes-duration on the index date were treated as a continuous variables. Household income was based on the 3-year average of equivalised disposable household income, converted to the corresponding percentile in the background population and treated as a continuous variable. Duration of residence was calculated from the date of first immigration to Denmark, and categorized as < *10 years*, *10—15 years*, *15—20 years*, and ≥ *20 years*, with native Danes categorized in the latter group. Employment status was categorized as *Employed* (including students), *Retired* or *Unemployed* based on data for the year prior to the index date.

Macrovascular complications were assessed by ICD-10 primary diagnosis codes and procedure codes of stroke, ischemic heart disease and peripheral arterial disease recorded in the NPR before the index date (see S1 File for specific codes). DKD was defined by a UACR ≥300 mg/g recorded in RLRR/CLIS between 1 October 2015 and 31 December 2017. Other types of microvascular complications were not evaluated.

## Statistical analysis

Distributions of covariates and outcomes was tabulated for each migrant group.

Relative risks (RR) and 95% confidence intervals were computed using Poisson regression with robust (generalized estimating equations, *sandwich*) variance estimates [20, 21]. In the analysis of T2D prevalence, RR of prevalent T2D was computed in a crude model and in two models with different levels of adjustment to explore the effects of clinical and socioeconomic factors on migrant risks. The first model adjusted for sex and age (model 1); the second model further adjusted for employment status, household income, duration of residence, and region of residence (model 2). Additionally, age-specific T2D prevalence in each migrant group was modeled using a binomial model with log-link and restricted cubic splines as a function of age.

Similarly, the RR of not receiving each type of guideline-level T2D care was computed in a crude model and in two adjusted models to visualize risk disparities in a clinical context separately from the fully-adjusted model. The first model (model 1) adjusted for clinical risk factors in diabetes that may influence decision-making when planning diabetes care with the patient (sex, age, diabetes duration, prevalent macrovascular complications and DKD). The fully-adjusted model (model 2) further adjusted for socioeconomic factors that may influence a patient's health behavior and healthcare service usage (employment status, household income, duration of residence, and region of residence). All models were computed with native Danes as the reference group, and continuous variables were input as natural splines with knots at each quintile of the T2D study population (in the analysis of T2D prevalence, quintiles were calculated within that study population).

In order to assess potential disparities at other cut-off values of HbA1c and LDL cholesterol, cumulative empirical distributions were plotted with 95% confidence intervals computed with Kolmogorov-Smirnov's *D*.

Analyses were performed using R version 4.1.3 (2022-03-10) [22] and R extension packages from the Comprehensive R Archive Network [23–29]. The paper was written using RMarkdown in RStudio version 2022.2.1.461 [30] with colorblind-friendly palettes [31]. Complete tables of regression coefficients are available in supplementary material as estimates of relative risk (S5 File) and risk difference (S7 File). Estimates from models inputting continuous variables as quintiles (instead of splines) for easier interpretation of these variables are also available (S6 & S8 Files).

### Ethics

This study was approved by Statistics Denmark and the Danish Health Data Authority. Source code and analysis plans are available on the Open Science Framework repository [32].

## Results

### The type 2 diabetes cohort

Among 3,864,528 individuals age 25—99 on 1 January 2018, 262,837 individuals (6.8%) with T2D were identified. Of those with T2D, 254,097 (96.7%) had a diabetes duration of at least 6 months on the index date and were included in the subsequent analyses of care quality.

In this T2D study population, native Danes and migrants from Europe shared similar background characteristics, which differed from the remaining migrant groups in most variables. Characteristics of migrant groups in the study population for T2D care are shown in Table 1 (see S2 File for characteristics of the study population for analysis of T2D prevalence). The difference was largest in terms of age and household income, as migrants from these groups were younger (median age in these groups ranged between 52 [IQR 47–59] years in the Somalia-group to 63 [55–70] years in the Former Yugoslavia-group vs. 69 [60–76] years in native Danes) and poorer (median household income percentile ranged between 12 [8–19] in the Somalia-group to 31 [17–49] in the Sri Lanka-group vs. 40 [25–66] in native Danes). The geographic distribution of migrant groups across Denmark was uneven, as most migrant groups were more likely to live within the Capital Region of Denmark, especially those in the Pakistan-group (proportion residing in the Capital Region ranged between 8% in the Sri Lanka-group to 92% in the Pakistan-group vs. 25.5% of native Danes). Overall patterns in sex, diabetes duration and complication prevalence were less clear, although migrants from Somalia had the shortest diabetes duration (7.8 years) and lowest prevalence of complications (macrovascular complications: 12.5%, DKD: 4.0%) of all groups.

Compared to native Danes, the risk of prevalent T2D was elevated in all migrants except the Europe group, regardless of model, and highest in the Sri Lanka and Pakistan groups, (crude RR 3.98, 95% CI: 3.82–4.14 and 3.63, 95% CI: 3.51–3.75, resp.). In the age-specific prevalences, the increased prevalence was discernible from the lowest ages. Fig 2 shows the age-specific prevalence of T2D in each migrant group, with crude and adjusted overall risks.

### Disparities in diabetes monitoring

The proportion of individuals with T2D who had not received diabetes monitoring within the guideline-recommended intervals, varied between the five types of monitoring assessed. In native Danes, the proportions without monitoring of HbA1c, LDL-C, diabetic nephropathy, diabetic retinopathy, and diabetic foot disease were 6.8%, 13.1%, 43.6%, 43.2%, and 57.0%, respectively.

Compared to native Danes, migrant groups had similar or higher crude RR in these indicators of diabetes monitoring; the only exception being the Sri Lanka-group, who had lower risks in all indicators except screening for diabetic foot disease (RRs from 0.64, 95% CI: 0.51–0.80 for HbA1c to 0.79, 95% CI: 0.74–0.84 for diabetic nephropathy). The Somalia-group consistently had the highest risks of not receiving monitoring (RRs from 1.29, 95% CI: 1.22–1.37 in screening for diabetic nephropathy to 1.62, 95% CI: 1.36–1.94 in monitoring of HbA1c). The greatest disparity between migrants and native Danes was found in screening for diabetic foot disease, where the risk of not receiving monitoring was elevated in all migrant groups and increased by more than one-third in most.

**Table 1. Characteristics of study population for type 2 diabetes care.**

| Characteristics by origin | | Denmark | Middle East | Europe | Turkey | F. Yugoslavia | Pakistan | Sri Lanka | Somalia | Vietnam |
|---|---|---|---|---|---|---|---|---|---|---|
| N (%) | 254,097 (100) | 225,750 (88.8) | 7,865 (3.1) | 6,396 (2.5) | 4,472 (1.8) | 3,509 (1.4) | 2,715 (1.1) | 1,653 (0.7) | 957 (0.4) | 780 (0.3) |
| Sex | Female (%) | 99,937 (44.3) | 3,062 (38.9) | 2,991 (46.8) | 2,277 (50.9) | 1,682 (47.9) | 1,225 (45.1) | 713 (43.1) | 412 (43.1) | 376 (48.2) |
| Age | Mean (SD) | 67.7 (12.4) | 59.5 (11.3) | 67.7 (12.4) | 59.3 (11.0) | 62.5 (11.3) | 60.9 (11.3) | 57.5 (10.2) | 53.9 (11.7) | 63.4 (12.1) |
| Diabetes duration (years) | Mean (SD) | 8.6 (5.9) | 8.5 (5.9) | 7.9 (5.7) | 8.4 (5.8) | 8.4 (5.8) | 9.7 (6.3) | 9.5 (6.2) | 7.8 (5.6) | 8.0 (5.6) |
| Macrovascular complications | N (%) | 67,486 (29.9) | 2,187 (27.8) | 1,893 (29.6) | 1,261 (28.2) | 1,096 (31.2) | 869 (32.0) | 381 (23.0) | 120 (12.5) | 153 (19.6) |
| Diabetic kidney disease | N (%) | 12,313 (5.5) | 444 (5.6) | 293 (4.6) | 249 (5.6) | 207 (5.9) | 137 (5.0) | 110 (6.7) | 38 (4.0) | 73 (9.4) |
| Employment | Employed | 57,829 (25.6) | 1,540 (19.6) | 1,621 (25.3) | 1,123 (25.1) | 577 (16.4) | 949 (35.0) | 591 (35.8) | 202 (21.1) | 220 (28.2) |
| | Retired | 137,953 (61.1) | 2,101 (26.7) | 3,922 (61.3) | 1,429 (32.0) | 1,422 (40.5) | 1,024 (37.7) | 319 (19.3) | 127 (13.3) | 295 (37.8) |
| | Unemployed | 29,968 (13.3) | 4,224 (53.7) | 853 (13.3) | 1,920 (42.9) | 1,510 (43.0) | 742 (27.3) | 743 (44.9) | 628 (65.6) | 265 (34.0) |
| Household income percentile | Mean (SD) | 45.9 (25.3) | 23.8 (19.4) | 41.4 (26.8) | 24.9 (18.5) | 28.2 (19.9) | 25.0 (21.1) | 34.5 (21.8) | 15.4 (11.9) | 28.3 (20.9) |
| Duration of residence (years) | < 10 | 0 (0.0) | 542 (6.9) | 560 (8.8) | 45 (1.0) | 37 (1.1) | 98 (3.6) | 23 (1.4) | 60 (6.3) | 15 (1.9) |
| | 10—15 | 0 (0.0) | 323 (4.1) | 361 (5.6) | 47 (1.1) | 81 (2.3) | 65 (2.4) | 25 (1.5) | 21 (2.2) | 12 (1.5) |
| | 15—20 | 0 (0.0) | 1,662 (21.1) | 313 (4.9) | 179 (4.0) | 250 (7.1) | 173 (6.4) | 103 (6.2) | 269 (28.1) | 40 (5.1) |
| | > 20 | 225,750 (100.0) | 5,338 (67.9) | 5,162 (80.7) | 4,201 (93.9) | 3,141 (89.5) | 2,379 (87.6) | 1,502 (90.9) | 607 (63.4) | 713 (91.4) |
| Region of residence | Capital | 57,455 (25.5) | 3,817 (48.5) | 2,398 (37.5) | 2,640 (59.0) | 1,529 (43.6) | 2,499 (92.0) | 137 (8.3) | 364 (38.0) | 134 (17.2) |
| | Central Denmark | 50,383 (22.3) | 1,598 (20.3) | 1,074 (16.8) | 659 (14.7) | 524 (14.9) | 45 (1.7) | 724 (43.8) | 282 (29.5) | 229 (29.4) |
| | North Denmark | 26,208 (11.6) | 339 (4.3) | 526 (8.2) | 34 (0.8) | 173 (4.9) | 7 (0.3) | 128 (7.7) | 69 (7.2) | 103 (13.2) |
| | South Denmark | 51,343 (22.7) | 1,220 (15.5) | 1,564 (24.5) | 454 (10.2) | 928 (26.4) | 94 (3.5) | 497 (30.1) | 192 (20.1) | 267 (34.2) |
| | Zealand | 40,361 (17.9) | 891 (11.3) | 834 (13.0) | 685 (15.3) | 355 (10.1) | 70 (2.6) | 167 (10.1) | 50 (5.2) | 47 (6.0) |
| No hemoglobin-A1c monitoring | N (%) | 15,396 (6.8) | 630 (8.0) | 576 (9.0) | 299 (6.7) | 229 (6.5) | 207 (7.6) | 72 (4.4) | 106 (11.1) | 74 (9.5) |
| No LDL-cholesterol monitoring | N (%) | 29,618 (13.1) | 1,204 (15.3) | 987 (15.4) | 690 (15.4) | 436 (12.4) | 430 (15.8) | 153 (9.3) | 176 (18.4) | 98 (12.6) |
| No diabetic nephropathy screening | N (%) | 98,461 (43.6) | 3,809 (48.4) | 3,175 (49.6) | 2,085 (46.6) | 1,584 (45.1) | 1,338 (49.3) | 569 (34.4) | 539 (56.3) | 321 (41.2) |
| No diabetic retinopathy screening | N (%) | 97,515 (43.2) | 3,978 (50.6) | 3,275 (51.2) | 2,216 (49.6) | 1,828 (52.1) | 1,600 (58.9) | 516 (31.2) | 535 (55.9) | 335 (42.9) |
| No diabetic foot disease screening | N (%) | 128,579 (57.0) | 6,414 (81.6) | 4,166 (65.1) | 3,606 (80.6) | 2,810 (80.1) | 2,177 (80.2) | 1,092 (66.1) | 836 (87.4) | 672 (86.2) |
| Hemoglobin-A1c level | Mean (SD) | 52.3 (13.5) | 55.9 (16.0) | 52.7 (13.9) | 58.3 (16.8) | 56.3 (15.4) | 57.8 (15.5) | 56.9 (14.4) | 58.1 (18.3) | 53.3 (13.3) |
| LDL-cholesterol level | Mean (SD) | 2.2 (0.9) | 2.2 (0.9) | 2.3 (1.0) | 2.2 (0.9) | 2.2 (0.9) | 2.2 (0.9) | 2.2 (0.9) | 2.6 (0.9) | 2.1 (0.9) |
| No glucose-lowering drugs | N (%) | 47,497 (21.0) | 1,391 (17.7) | 1,534 (24.0) | 567 (12.7) | 510 (14.5) | 449 (16.5) | 182 (11.0) | 182 (19.0) | 122 (15.6) |
| No lipid-lowering drugs | N (%) | 71,717 (31.8) | 2,722 (34.6) | 2,408 (37.6) | 1,317 (29.4) | 909 (25.9) | 882 (32.5) | 445 (26.9) | 548 (57.3) | 219 (28.1) |
| No ACEI/ARB | N (%) | 81,184 (36.0) | 4,074 (51.8) | 2,620 (41.0) | 2,130 (47.6) | 1,395 (39.8) | 1,345 (49.5) | 809 (48.9) | 634 (66.2) | 372 (47.7) |
| No antiplatelet therapy | N (%) | 145,712 (64.5) | 5,398 (68.6) | 4,232 (66.2) | 3,033 (67.8) | 2,334 (66.5) | 1,724 (63.5) | 1,197 (72.4) | 852 (89.0) | 590 (75.6) |

F. Yugoslavia, Former Yugoslavia; LDL-cholesterol, low-density lipoprotein cholesterol; ACEI/ARB, angiotensin-converting enzyme-inhibitors or angiotensin receptor blockers

Risk estimates were stable across models in the Europe-group, but attenuated with increasing adjustment in other migrant groups. Fig 3 shows the crude and adjusted risks of not receiving each type of monitoring in migrants compared to native Danes.

## Disparities in biomarker levels

In the T2D study population, 250,475 (98.6%) had at least one measurement of HbA1c, and 248,813 (97.9%) had a measurement of LDL-C and were included in the respective analyses of biomarker levels.

The overall mean of HbA1c levels was 52.7 mmol/mol, and 37.2% of native Danes had an HbA1c level $\geq$ 53 mmol/mol. Compared to native Danes, the crude risk of having an HbA1c level $\geq$ 53 mmol/mol was increased in all migrant groups except the Europe and Vietnam groups (RRs from 1.27, 95% CI: 1.24–1.30, in the Middle East-group to 1.46, 95% CI: 1.41–1.52, in the Pakistan-group).

The overall mean of LDL-C levels was 2.2 mmol/L, and 28.3% of native Danes had an LDL-C level $\geq$ 2.6 mmol/L. The crude risk of having an LDL-C $\geq$ 2.6 mmol/L was increased

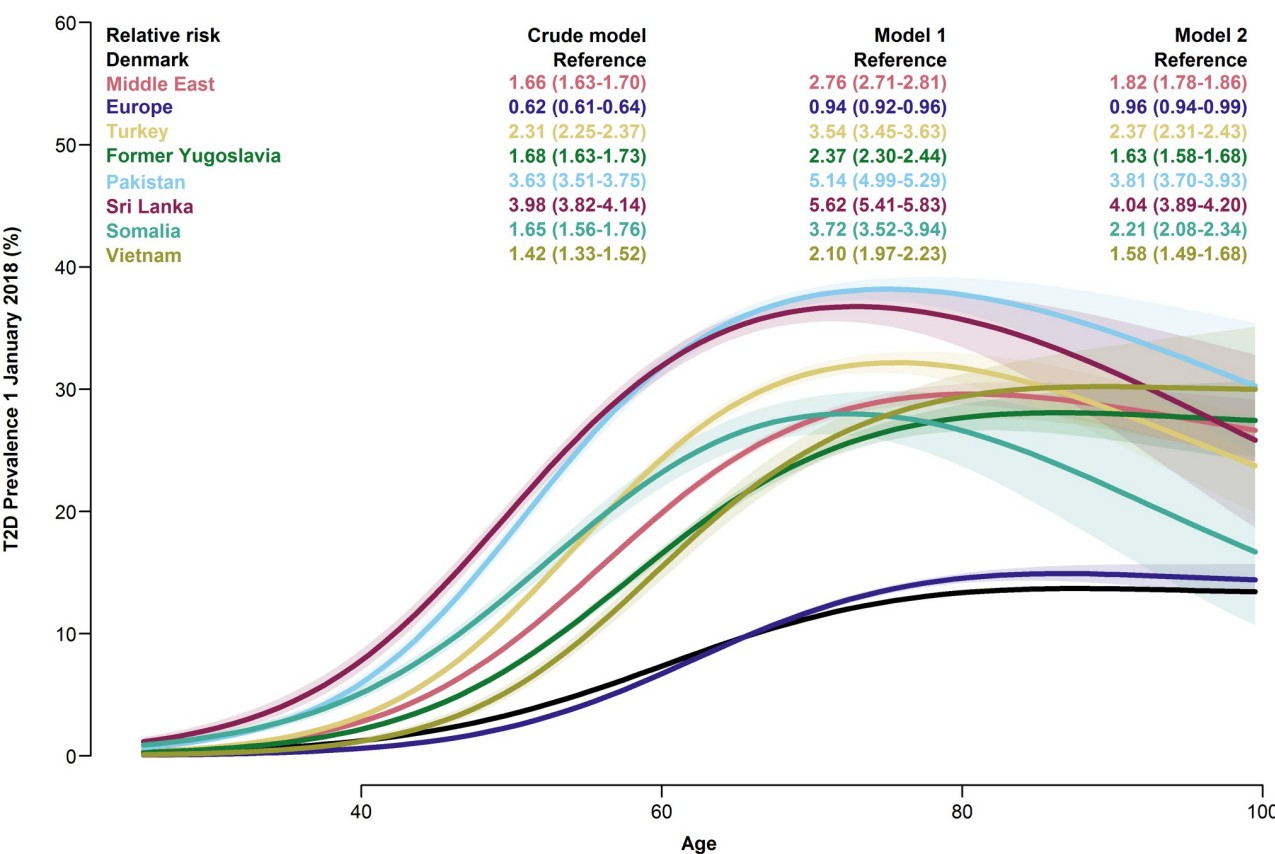

**Fig 2. Prevalence of type 2 diabetes.** Age-specific prevalences of type 2 diabetes with 95% CIs for each migrant group, and overall relative risks with 95% CIs from three models. Model 1: adjusted for age and sex. Model 2: adjusted for age, sex, employment status, household income, duration of residence and region of residence.

in all migrant groups except the Sri Lanka and Vietnam groups (RRs from 1.08, 95% CI: 1.03–1.14, in the Former Yugoslavia-group to 1.78, 95% CI: 1.67–1.90, in the Somalia-group).

For both HbA1c and LDL-C, these disparities appeared consistent across a range of alternative, clinically relevant target values in the empirical cumulative distributions. Adjustments in model 1 roughly halved the size of the increased risks compared to crude estimates, while further adjustment for socioeconomic factors in model 2 only had minor effect on the estimates compared to model 1.

Fig 4 shows the empirical cumulative distribution of HbA1c (panel A) and LDL-C (panel B) for each migrant group, and the crude and adjusted risks of exceeding the corresponding target level compared to native Danes.

## Disparities in pharmacological treatment

The subgroups of the population, in whom CG recommended pharmacological treatment, varied between the four types of drugs assessed, as did the proportion of native Danes who did not receive guideline-recommended care. For treatment with GLD, LLD, ACEI/ARB, and APT, the size of the subgroups and the corresponding proportion of native Danes without treatment in each group were 140,208 (6.9%), 128,707 (34.9%), 77,414 (28.4%), and 77,414 (34.8%), respectively (see S2 File for characteristics of these groups).

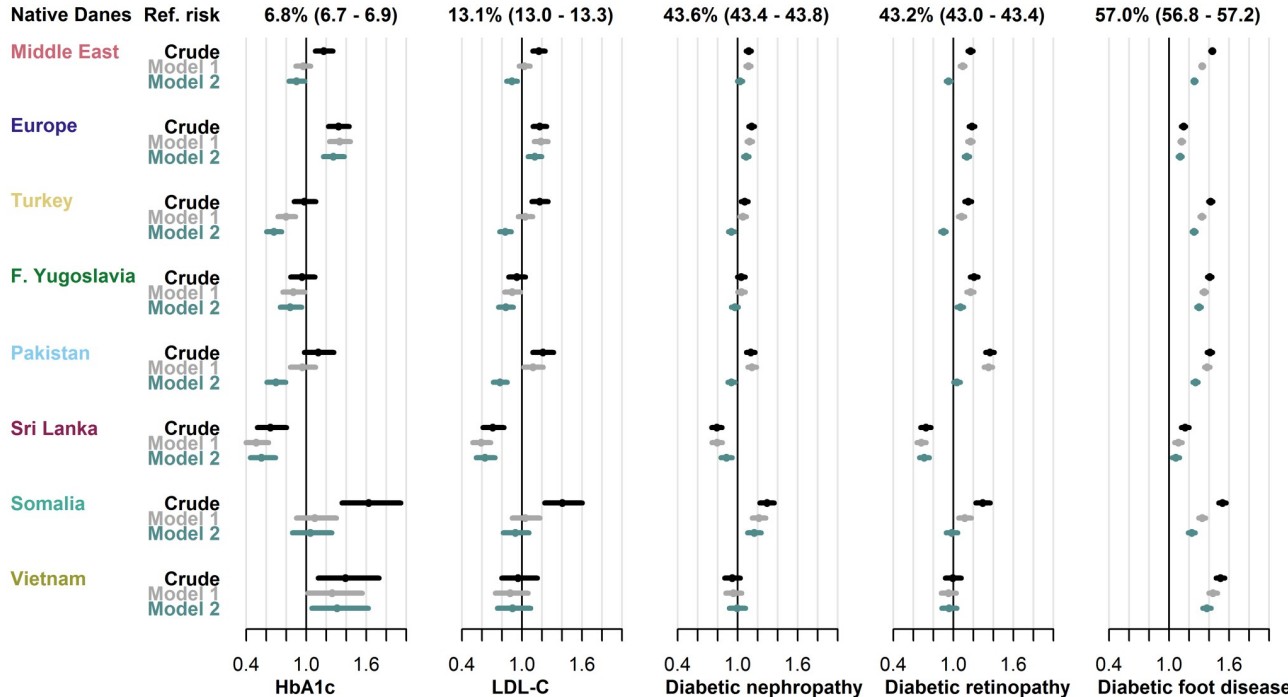

**Fig 3. Risk of not receiving guideline-recommended type 2 diabetes monitoring.** Relative risks in each migrant group with 95% CIs from three models across five types of monitoring. Model 1: adjusted for age, sex, diabetes duration and prevalent complications. Model 2: adjusted for age, sex, diabetes duration, prevalent complications, employment status, household income, duration of residence and region of residence. F. Yugoslavia; Former Yugoslavia; HbA1c, hemoglobin-A1c; LDL-C, low-density lipoprotein cholesterol.

Compared to native Danes, the crude risk of not receiving GLD was increased in the Europe (RR 1.45, 95% CI: 1.31–1.60) and Somalia (RR 1.67, 95% CI: 1.34–2.09) groups. Both groups also had increased risk of not receiving LLD (RR 1.14, 95% CI: 1.09–1.19) in Europe-group and 1.64, 95% CI: 1.52–1.77 in the Somalia-group). In the remaining migrant groups, the risk of not receiving GLD and LLD was similar or lower than in native Danes.

While the risk of not receiving ACEI/ARB was higher in five of the eight migrant groups (RRs from 1.24, 95% CI: 1.15–1.34 in the Turkey-group to 1.50, 95% CI: 1.25–1.82 in the Somalia-group), only the Somalia-group was at higher risk of not receiving APT than native Danes (RR 1.53, 95% CI: 1.31–1.78).

The RR estimates for GLD varied between the models in some groups, particularly in the Pakistan-group, while the estimates for LLD, ACEI/ARB and APT were stable. Although the RR of not receiving LLD in the Somalia-group attenuated substantially following adjustment, it remained elevated. Fig 5 shows the crude and adjusted risks of not receiving each type of pharmacological treatment in migrants compared to native Danes.

## Discussion

This study provides information on prevalence of T2D and disparities between migrant groups and native Danes across eleven indicators of guideline-recommendations on monitoring, bio-marker levels and pharmacological treatment, using population-wide data.

We found that the prevalence of T2D was higher in migrants compared to native Danes, especially in migrants from Sri Lanka and Pakistan, who were four to five times more likely to have T2D compared to native Danes.

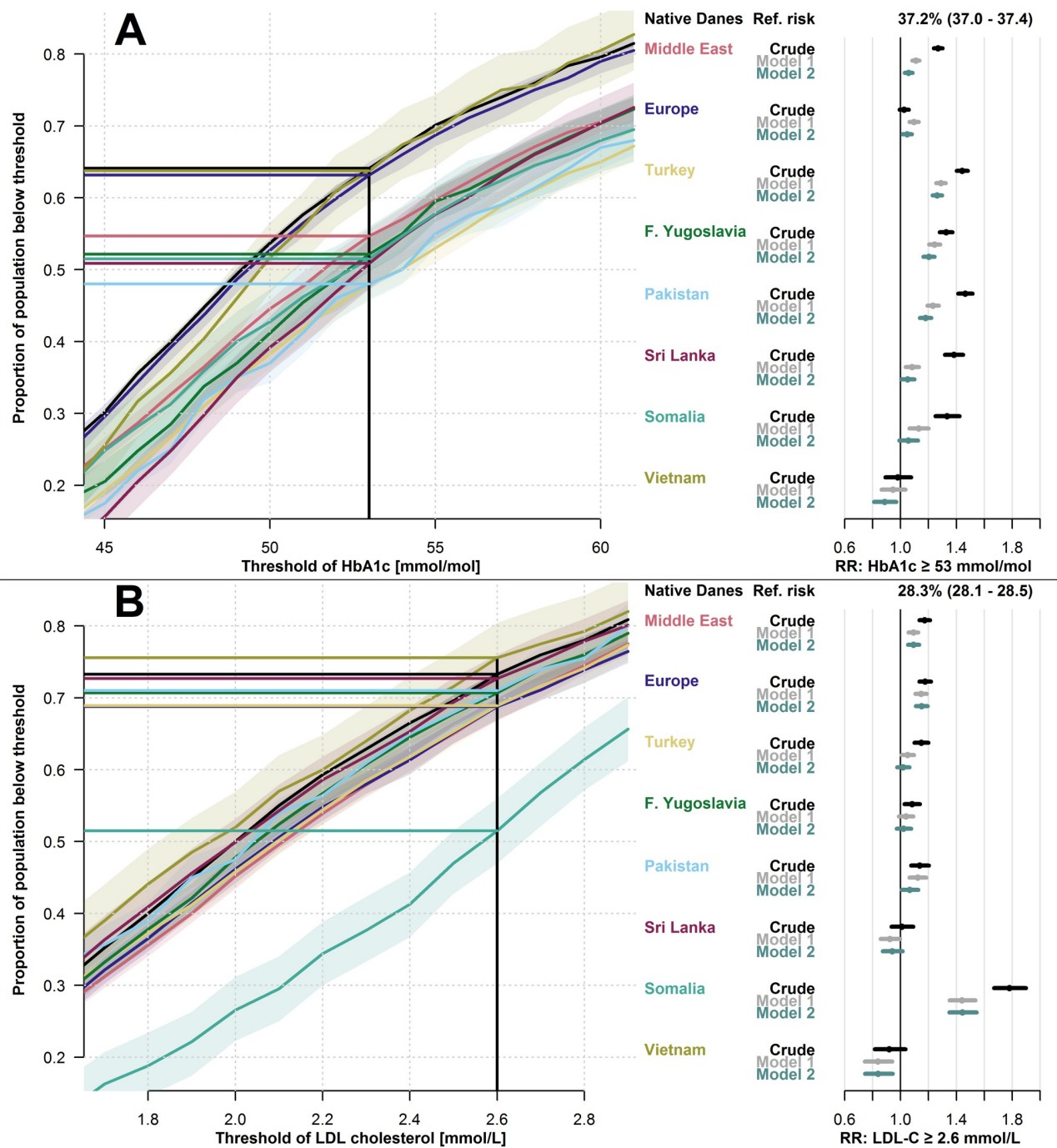

**Fig 4. Biomarker levels in type 2 diabetes and risk of exceeding guideline recommendations.** A: Empirical cumulative distribution of hemoglobin-A1c (HbA1c) with 95% CIs and relative risk of exceeding 53 mmol/mol from three models. B: Empirical cumulative distribution of low-density lipoprotein cholesterol (LDL-C) with 95% CIs and relative risk of exceeding 2.6 mmol/L from three models. Model 1: adjusted for age, sex, diabetes duration and prevalent complications. Model 2: adjusted for age, sex, diabetes duration, prevalent complications, employment status, household income, duration of residence and region of residence. F. Yugoslavia; Former Yugoslavia.

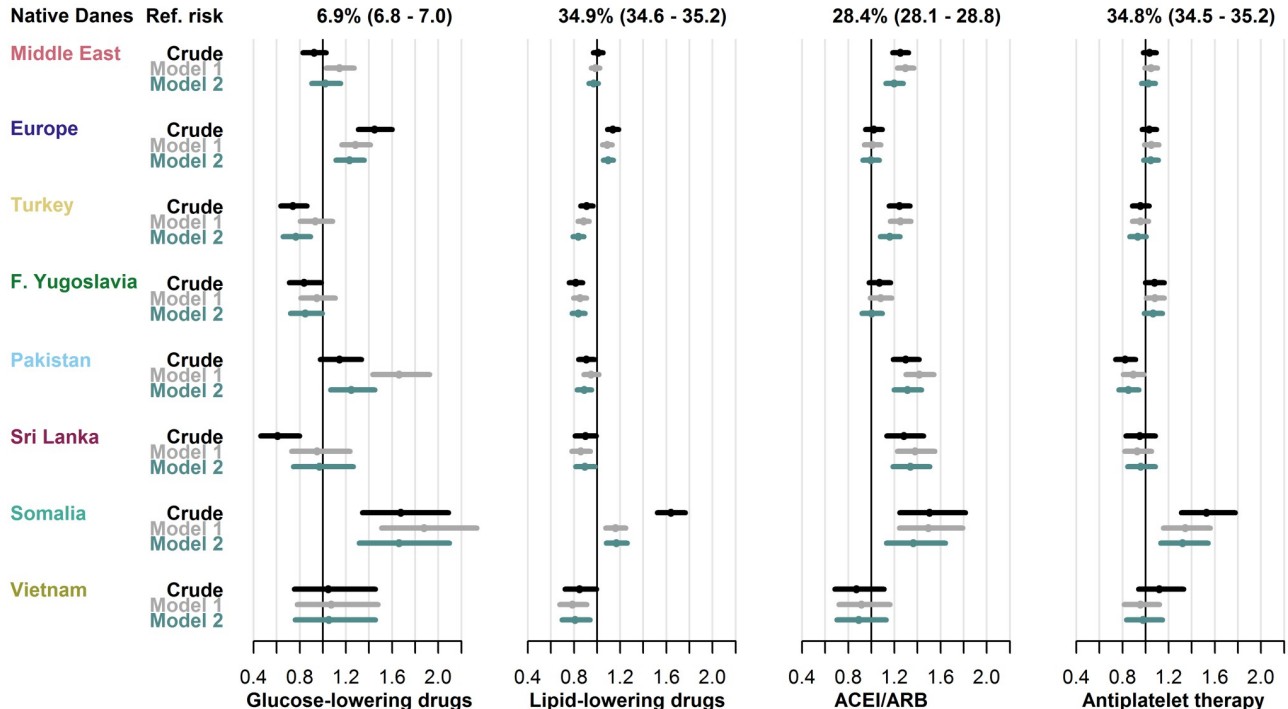

**Fig 5. Risk of not receiving guideline-recommended pharmacological treatment in type 2 diabetes.** Relative risks in each migrant group with 95% CIs from three models across four types of drugs. Model 1: adjusted for age, sex, diabetes duration and prevalent complications. Model 2: adjusted for age, sex, diabetes duration, prevalent complications, employment status, household income, duration of residence and region of residence. GLD, glucose-lowering drugs; LLD, lipid-lowering drugs; ACEI/ARB, angiotensin-converting enzyme-inhibitors or angiotensin receptor blockers; F. Yugoslavia; Former Yugoslavia; HbA1c, hemoglobin-A1c; LDL-C, low-density lipoprotein cholesterol.

Failure to fulfill guideline recommendations for T2D care was common, as the proportion of native Danes unable to meet recommendations was below 10% in only two indicators (HbA1c-monitoring and GLD-treatment), and above 25% in eight (all other indicators except monitoring of LDL-C). Failure was more common in migrants, particularly in terms of monitoring and control of HbA1c and LDL-C levels. In the majority of these indicators, we observed increased crude risks at a significance level of $\alpha = 0.05$ in most migrant groups (see S10 File), while no clear overall pattern across migrant groups was observed in the indicators of pharmacological treatment. The risk varied between migrant groups, as migrants from Sri Lanka were more likely than native Danes to meet recommendations for monitoring, while migrants from Somalia had increased crude risk in all eleven indicators, and the highest risk of all migrants in nine. Disparity between migrants and native Danes was largest in the indicators of glycemic control and screening for diabetic foot disease, where crude risk was increased by more than one-third in most migrant groups, and in the indicator of lipid control, where migrants from Somalia had almost twice the risk of native Danes.

These patterns of increased risk among migrants persisted after adjusting for differences in clinical risk factor profiles (sex, age, diabetes duration, complication status) that may have influenced clinical decision-making. Following further adjustment for socioeconomic factors (employment status, household income, region and duration of residence) the pattern of increased risk persisted in the indicators of biomarker control, but was not clear in the indicators of monitoring and pharmacological treatment, although migrants from Europe and

Somalia remained at increased risk in the majority of all indicators. In the majority of indicators, the attenuation from crude to adjusted estimates was mainly due to adjustment for young age, low income, and residence in the Capital Region of Denmark, as these variables were risk factors of non-fulfillment in nearly all indicators and were much more common among migrants than native Danes.

## Type 2 diabetes prevalence

T2D was more prevalent in all migrant groups except those from Europe. While similar findings have previously been reported in Danish studies [1, 4] and across Europe [2], our study shows this risk persisting after adjusting for differences in socioeconomic position in a Danish setting. A previous study in Aarhus municipality reported an age-standardized prevalence 5.5 times higher in migrants from Turkey relative to native Danes [4]. While the RR in our study was not as high, the difference may be explained by different age-ranges in the study populations and a higher prevalence among native Danes in our study (6.6% vs. 3.6%).

## Monitoring

Our findings of lower likelihood of T2D monitoring in most migrant groups contrast findings from prior studies, where migrants received similar or more monitoring compared to native Danes [4], Norwegians [5] and Swedes [7], but is in line with recent findings of inferior monitoring in ethnic minority groups in England [8].

While HbA1c, LDL-C and UACR can be sampled at the GP, screening for diabetic retinopathy and diabetic foot disease is performed by an ophthalmologist or a podiatrist at an external location, and migrants may be particularly vulnerable to these services not being integrated locally. However, the proportion of the T2D population without timely monitoring varied substantially between the types of monitoring performed at the GP, suggesting that this distinction may be of less importance. Socioeconomic factors such as household income and region of residence were risk factors in all types of monitoring, and appear to contribute substantially to the observed disparities in migrants, despite diabetes monitoring being provided free of charge in all regions of Denmark. Screening for diabetic foot disease is the only service where patient expenditures are not fully covered by the public health insurance, which may explain why it was the indicator with the highest proportion of individuals without timely monitoring. Furthermore, the large disparity between migrants and native Danes in this indicator, which persisted after adjusting for socioeconomic variables, suggests that economic barriers may disproportionately limit access to care in migrant groups. However, the disparities observed in the freely offered monitoring services suggest that the monitoring of diabetes is affected by socioeconomic status beyond the patient's direct expenses.

Migrants tend to have more healthcare contacts than native Danes [33], and may thus be more likely to receive testing for HbA1c, LDL-C and UACR that does not represent routine diabetes monitoring. While we expect this effect to be minor, it could have biased estimates towards overestimating the quality of monitoring in migrants.

Although CG do not specify how urgently baseline-screening should be performed in the newly-diagnosed, we considered a cut-off at 6 months after diagnosis a reasonable time-frame to evaluate monitoring of the whole population, although the quality of baseline-screening may differ from subsequent routine monitoring. In sensitivity analyses of baseline-monitoring compared to routine monitoring (see S4 File), the reference risk differed between individuals with diabetes onset before the start of the monitoring-window compared to those with a diabetes duration shorter than the monitoring window (individuals with shorter duration were more likely to have received monitoring of HbA1c and LDL-C, and less likely to have received

other types of monitoring), but this did not appear to bias our estimates of RRs in migrants substantially.

## Biomarker levels

In line with previous studies, most migrant groups were less likely than native Danes to achieve glycemic control [5–8]. The magnitude of the risk was similar to what has been reported in previous studies, although it may seem lower when compared to estimates reported as odds ratios (OR) in other studies, since OR lead to exaggerated risk estimates if interpreted as RR when the outcome is common [34]; e.g. a study in Scotland reported a crude OR of dysglycemia of 2.2 in the Pakistani group, and a dysglycemia-prevalence of roughly 50% in the reference group [6], which corresponds to a RR of 1.4 [35], similar to our estimate.

Few studies have reported on cholesterol levels. A small study in the US found a trend towards higher prevalence of dyslipidemia in Somali migrants [36], although much weaker than the RR found in our study. This may be explained by the higher LDL-C levels in the reference group of that study (proportion with LDL-C $\geq$ 2.6 mmol/mol: 39% vs. 28% in our study). Our findings of similar LDL-C levels in native Danes and migrants from Vietnam and Sri Lanka may correspond to the findings of similar cholesterol levels in ethnic Europeans and ethnic South Asians in England [10], although the differences in categories do not allow direct comparisons.

To the best of our knowledge, the increased LDL-C levels among migrants in our study have not been reported previously. Several studies have examined obesity in migrants from Somalia and found high prevalences of obesity and increased waist-to-hip ratio in women, but not in men [37–39]. If this pattern between sexes extends to migrants from Somalia in Denmark, it seems unlikely that anthropometric factors can explain the very high levels of LDL-C in this population, as the risk remained unchanged in analyses stratified by sex (see analysis of LDL-C levels in S3 File).

## Pharmacological treatment

Most studies of pharmacological treatment in T2D have analysed the whole population, rather than subgroups with a clinical indication for treatment, and have evaluated incident medication use in the newly-diagnosed, making direct comparisons to our study difficult. While our finding of lower or similar risk of GLD non-use in most migrant groups is in line with previous studies comparing migrants to native populations [5, 7], the lower risk of LLD non-use in most migrant groups contrast findings from prior studies. A study from Denmark found that migrants from Turkey, Pakistan and the Former Yugoslavia newly-diagnosed with T2D were less likely to initiate statin therapy than native Danes [40], while a study in Norway found lower proportions of statin users in migrants than in natives with T2D [5], and in the UK, individuals from ethnic minorities with incident T2D and a guideline-indication for treatment had lower rates of LLD prescribing than people of European ethnicity [10]. Some of these differences may be explained by a longer time to initiation of LLD treatment in migrant groups, which would have a greater negative effect on associations in studies of incident T2D than in our cross-sectional design.

The proportion of non-users was much lower in the indicator of GLD compared to the other types of drugs assesed in our study, perhaps indicating a higher focus among physicians and patients on improving HbA1c levels than improving other risk factors in T2D.

In exploratory analyses, we observed a higher risk of non-use of LLD among individuals with an indication for treatment based on elevated LDL-C levels and age alone compared to those with an indication due to prevalent complications (see analysis of lipid-lowering drugs

in S5 File), which was also apparent in the proportion of LLD users in the two groups (40.9% vs. 81.1%). Although some of this difference can be attributed to our study design, which defines some of the study population based on elevated LDL-C levels (LLD use among individuals with concurrently elevated LDL-C levels is less likely due to the lipid-lowering effect of these drugs), there appears to be a particular need to increase the use of LLD as primary prophylaxis in the T2D population due to high levels of LDL-C.

The overall proportion of LLD users in our study (69.4%) was substantially higher than the 55% reported in a recent Danish study of individuals with T2D [41]. A stricter outcome measure was used in that study, however, as only individuals with continuous LLD coverage throughout the year were defined as users, and the difference may represent lapses in adherence to LLD in our study population.

The increased risk of dysglycemia in the Somalia-group coincided with an increased risk of not receiving GLD, and this pattern was also observed for dyslipidemia and LLD in the Somalia and Europe groups, indicating a particular need to increase uptake of pharmacological treatment in these groups.

Few studies have reported on ACEI/ARB and APT in migrants, but a recent study from Italy found lower odds of treatment in most migrant groups [42]. While we also found ACEI/ARB treatment less likely in most migrants, the risk of not receiving APT was similar in most migrants and native Danes in our study, with the exception of migrants from Somalia, who had higher risk of not receiving treatment. As the indications for treatment with ACEI/ARB and APT were identical in our study, one might expect similar risk patterns in these outcomes, but this was not the case, perhaps due to differences between groups in terms of other indications for treatment, contraindications, awareness and attitude to these drugs.

In all types of pharmacological treatment, age above 79 was associated with a sharp increase in the risk of non-use (see S6 File), suggesting that deprescribing in the elderly with T2D, recently described for GLD in a Danish study [43], also extends to LLD, ACEI/ARB and APT despite clinical indications for treatment.

## Strengths and limitations

The unique strength of this study was the large study population with data on an entire nation's socio-demography, healthcare services, biomarker levels and pharmacological treatment, which maximizes the internal validity of our findings. The quality of the register data is likely to be very high, as it is captured automatically and used for civil registration and services, or billing and reimbursement purposes within the public healthcare system. The biomarker data is also captured automatically in the clinical databases where the data for this study was extracted from, and is likely to be accurate, although the RLRR is an emerging data resource. The RLRR is compiled from several regional laboratory databases and its contents have not been validated, but our initial concerns over data completeness across regions were relieved by the low inter-regional variation in the time-frame of this study (see S9 File). While hospital diagnoses are susceptible to potential clinician error at the time of recording, diagnoses of cardiovascular disease in the register data have previously been validated and found highly accurate [44].

The close alignment of outcomes to clinical guideline-recommendations provides specific and clinically relevant knowledge on areas of care with potential for improvement, and the use of relative risk to communicate risks makes interpretation intuitive (see S7 & S8 Files. for absolute risk differences). The ability of this study to provide risk estimates in migrant groups with a high representation in Denmark makes the results particularly relevant in a Danish context and in countries with similar migrant populations.

There were several limitations to this study. The evaluation of quality of care using binary process and outcome indicators does not capture qualitative differences in the provided care, nor do they account for valid patient-specific reasons for not providing care or achieving biomarker control. In some individuals, failure to achieve biomarker control may be unrelated to the quality of provided care (e.g. due to genetic factors), and a lack of monitoring may not affect risk factors and the risk of adverse outcomes.

Migrant origin, socioeconomic position, lifestyle and healthcare utilization are intrinsically linked beyond the dimensions covered in our data. As the focus of this study was on risk rather than causality, we did not adjust for possible differences in GP-behavior. As this study maps the risks of sub-optimal care quality in migrant groups, the reported disparities cannot be interpreted as causal effects. The observed disparities between migrants and native Danes diminished considerably after adjustment for differences in clinical and socioeconomic characteristics, and could to some extent be attributed to a much higher prevalence of risk factors of poor T2D care in migrants; particularly young age, low household income and residence in the Capital Region. However, the adjusted risks in migrants should be interpreted with caution due to the extremely skewed distributions of risk factors between migrants and native Danes. Ultimately, the three models provide distinct perspectives on the risks in migrants and highlight the influence of the clinical and socioeconomic contexts in these risks.

Furthermore, there were limits in the data that prevented evaluation of additional indications of pharmacological treatment, including neuropathy, retinopathy, hypertension, smoking habits, obesity and family history of complications. Also, the historic window of laboratory data was limited, and DKD prevalence may be underestimated due to cases with onset before 2015 without subsequent UACR samples.

Finally, it is important to note that comparisons between studies of T2D care should be made with caution, as quality of care has improved substantially over the last two decades [8, 45, 46], and differences in care quality may simply reflect temporal differences between studies. Also, differences in risk estimates between studies may reflect differences in reference group risk, rather than differences in the absolute risk of migrants. While overall risks in most indicators of T2D care appear similar to reports from other countries, levels of HbA1c and LDL-C are lower in the Danish T2D population [8, 45], and the relative risks of glycemic and lipid control in migrants reported here may not extend to migrants with T2D in other settings.

## Conclusions

In Denmark, T2D care quality below guideline-recommended standards was common, with room for substantial improvement in most aspects of monitoring, biomarker levels and pharmacological treatment. Compared to native Danes, prevalence of T2D was higher in most migrant groups, quality of monitoring was inferior, and most migrant groups had higher risks of not achieving glycemic and lipid control, despite similar quality of pharmacological treatment. Care was poorest in migrants from Somalia, who had increased risk in all of the eleven indicators assessed, and the highest risk of all migrant groups in nine of these, including a nearly double risk of dyslipidemia compared to native Danes. To some extent, the disparities between migrants and native Danes could be attributed to a much higher prevalence of risk factors of poor T2D care in migrants; specifically young age, low household income and residence in the Capital Region. Health planners and general practitioners in Denmark should focus on improving the areas of insufficient T2D care highlighted in this study, and further studies should work to identify effective interventions.

## Supporting information

**S1 File. Register codes used to define variables.**
(HTML)

**S2 File. Characteristics of study populations for analysis of T2D prevalence and pharmacological treatment.**
(HTML)

**S3 File. Supplementary analyses stratified by sex.**
(HTML)

**S4 File. Supplementary analyses of monitoring stratified by diabetes duration.**
(HTML)

**S5 File. Regression coefficients of relative risk from all analyses.**
(HTML)

**S6 File. Regression coefficients of relative risk using quintiles instead of splines.**
(HTML)

**S7 File. Regression coefficients of absolute risk difference.**
(HTML)

**S8 File. Regression coefficients of absolute risk difference using quintiles instead of splines.**
(HTML)

**S9 File. Biomarker data coverage in the background population across geographic regions of Denmark.**
(HTML)

**S10 File. Overview of risks in migrants compared to native Danes.**
(PDF)

## Author Contributions

**Conceptualization:** Anders Aasted Isaksen, Annelli Sandbæk, Mette Vinther Skriver, Gregers Stig Andersen, Lasse Bjerg.

**Data curation:** Anders Aasted Isaksen.

**Formal analysis:** Anders Aasted Isaksen.

**Funding acquisition:** Anders Aasted Isaksen, Annelli Sandbæk.

**Methodology:** Anders Aasted Isaksen, Annelli Sandbæk, Gregers Stig Andersen, Lasse Bjerg.

**Project administration:** Anders Aasted Isaksen.

**Supervision:** Annelli Sandbæk, Mette Vinther Skriver, Gregers Stig Andersen, Lasse Bjerg.

**Validation:** Lasse Bjerg.

**Visualization:** Anders Aasted Isaksen.

**Writing – original draft:** Anders Aasted Isaksen.

**Writing – review & editing:** Annelli Sandbæk, Mette Vinther Skriver, Gregers Stig Andersen, Lasse Bjerg.

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
