## [Decision Letter · Decision Letter 0]

13 Dec 2022

PGPH-D-22-01596

Guideline-level monitoring, biomarker levels and pharmacological treatment in migrants and native Danes with type 2 diabetes: population-wide analyses

Dear Dr. Isaksen,

Thank you for submitting your manuscript to PLOS Global Public Health. After careful consideration, we feel that it has merit but does not fully meet PLOS Global Public Health’s publication criteria as it currently stands. Therefore, we invite you to submit a revised version of the manuscript that addresses the points raised during the review process.

Please address all the issues raised by the reviewers. 

We look forward to receiving your revised manuscript.

Kind regards,

Palash Chandra Banik, MPhil

Academic Editor

Journal Requirements:

1. We ask that a manuscript source file is provided at Revision. Please upload your manuscript file as a .doc, .docx, .rtf or .tex.

2. We have noticed that you have uploaded Supporting Information files, but you have not included a list of legends. Please add a full list of legends for your Supporting Information files after the references list. 

3. Since your data is not available for proprietary reasons, please explain via email why the data is not available. Please also include the contact information for the third party organization that should be contacted should other researchers want to request access to this data and please include the full citation of where the data can be found. We also request that you verify with us via email that any researcher will be able to obtain the data set in the same manner that the you have obtained it. If you feel you are unwilling or unable to adhere to this policy, please explain your reasons by return email and your exemption request will be escalated to the editor for approval. Your exemption request will be handled independently and will not hold up the peer review process, but will need to be resolved should your manuscript be accepted for publication. One of the Editorial team will be in touch if they require more information.

Additional Editor Comments (if provided):

Reviewers' comments:

Reviewer's Responses to Questions

**Comments to the Author**

1. Does this manuscript meet PLOS Global Public Health’s publication criteria? Is the manuscript technically sound, and do the data support the conclusions? The manuscript must describe methodologically and ethically rigorous research with conclusions that are appropriately drawn based on the data presented.

Reviewer #1: Yes

Reviewer #2: Partly

2. Has the statistical analysis been performed appropriately and rigorously?

Reviewer #1: Yes

Reviewer #2: No

3. Have the authors made all data underlying the findings in their manuscript fully available (please refer to the Data Availability Statement at the start of the manuscript PDF file)?

Reviewer #1: Yes

Reviewer #2: Yes

4. Is the manuscript presented in an intelligible fashion and written in standard English?

Reviewer #1: Yes

Reviewer #2: Yes

5. Review Comments to the Author

Reviewer #1: It is a good work. I have only a comment on the abstract section that missed the result part.

The authors should include result part under the abstract.

Additionally, I am not comfortable with the text indentation and Alignment. Alignment should be Justified.

Tables should be beneath its label (Table 3).

Reviewer #2: An important study assessing the disparities in prevalence, monitoring, and implementation of clinical guidelines for type 2 diabetes comparing native Danes and immigrants using register-based/secondary data.

The justification to exclude adults younger than 25 years, second-generation descendants of immigrants or immigrants residing in Denmark for less than 3 years on 50 the index date, and migrants from countries not in the selected origin categories, is not provided. Please clarify. Figure 1 was not legible/clear which means that I wasn’t able to read it.

The authors state that they used datasets from various sources, which is commendable. However, information about how the data linking was achieved at the individual level is missing. Was there incomplete data linkage or did the investigators find challenges associated with data linkage such as inaccurate data linkage or missing unique identifiers on these datasets? Besides, I also observe that records were available from 5 different sources available at different times, for instance, data on diabetes-specific services are available from 1990 while biomarkers data were only available from 2015 onward. How you addressed potential biases associated with incomplete data linkage in your context requires clarification.

About the definition of diabetes based on the 4 events (lines 86-94). Consider event number (iv), the purchase of GLD drugs. Was the purchaser always a diabetes patient? Or somebody else [such as a family member/partner/friend] may have purchased the drugs for the patient? There is a need to clarify this assumption in the Danish context to prevent misclassification for diabetes, potentially overestimating the prevalence.

ATC abbreviation is not defined at the point of initial mention making the reading difficult.

While ethical approval is not required for register-based studies in Denmark, are there any implications concerning the use of patient data for this study? What about the privacy and confidentiality of patient information? Surprisingly, there are no ethical considerations for use of patient data for such types of studies in Denmark!

Regarding the adjusted models, what model selection approach did you use to identify the model that best fits your data? This information should be provided.

Model identification and selection. What necessitated the development of more than 1 adjusted model in each sub-analysis? Is there a statistical approach that can be used to find a single parsimonious adjusted model that best fits your dataset in each sub-analysis? I find it unnecessary to develop more than one adjusted model when you can develop a single parsimonious model that best fits your data to be compared with the crude model. This applies to figures 3, 4, and 5.

The discussion is well written but may change depending on how the authors address the comments raised on the methods and results.

6. PLOS authors have the option to publish the peer review history of their article (what does this mean?). If published, this will include your full peer review and any attached files.

**Do you want your identity to be public for this peer review?** For information about this choice, including consent withdrawal, please see our Privacy Policy.

Reviewer #1: No

Reviewer #2: No

---

## [Decision Letter · Decision Letter 1]

8 Feb 2023

PGPH-D-22-01596R1

Guideline-level monitoring, biomarker levels and pharmacological treatment in migrants and native Danes with type 2 diabetes: population-wide analyses

Dear Dr. Isaksen,

Thank you for submitting your manuscript to PLOS Global Public Health. After careful consideration, we feel that it has merit but does not fully meet PLOS Global Public Health’s publication criteria as it currently stands. Therefore, we invite you to submit a revised version of the manuscript that addresses the points raised during the review process.

EDITOR: Please insert comments here and delete this placeholder text when finished. Be sure to:

Indicate which changes you require for acceptance versus which changes you recommendAddress any conflicts between the reviews so that it's clear which advice the authors should followProvide specific feedback from your evaluation of the manuscript

Please ensure that your decision is justified on PLOS Global Public Health’s publication criteria and not, for example, on novelty or perceived impact.

We look forward to receiving your revised manuscript.

Kind regards,

Palash Chandra Banik, MPhil

Academic Editor

Journal Requirements:

Additional Editor Comments (if provided):

Reviewers' comments:

Reviewer's Responses to Questions

**Comments to the Author**

1. If the authors have adequately addressed your comments raised in a previous round of review and you feel that this manuscript is now acceptable for publication, you may indicate that here to bypass the “Comments to the Author” section, enter your conflict of interest statement in the “Confidential to Editor” section, and submit your "Accept" recommendation.

Reviewer #2: All comments have been addressed

Reviewer #3: All comments have been addressed

Reviewer #4: All comments have been addressed

Reviewer #5: (No Response)

Reviewer #6: All comments have been addressed

2. Does this manuscript meet PLOS Global Public Health’s publication criteria? Is the manuscript technically sound, and do the data support the conclusions? The manuscript must describe methodologically and ethically rigorous research with conclusions that are appropriately drawn based on the data presented.

Reviewer #2: Yes

Reviewer #3: Yes

Reviewer #4: Yes

Reviewer #5: Yes

Reviewer #6: Yes

3. Has the statistical analysis been performed appropriately and rigorously?

Reviewer #2: Yes

Reviewer #3: Yes

Reviewer #4: Yes

Reviewer #5: Yes

Reviewer #6: Yes

4. Have the authors made all data underlying the findings in their manuscript fully available (please refer to the Data Availability Statement at the start of the manuscript PDF file)?

Reviewer #2: Yes

Reviewer #3: Yes

Reviewer #4: Yes

Reviewer #5: Yes

Reviewer #6: Yes

5. Is the manuscript presented in an intelligible fashion and written in standard English?

Reviewer #2: Yes

Reviewer #3: Yes

Reviewer #4: (No Response)

Reviewer #5: Yes

Reviewer #6: Yes

6. Review Comments to the Author

Reviewer #2: (No Response)

Reviewer #3: General Comment: This paper address inadequate care of diabetic patients among immigrates as compared to the native people and clearly showed gas which give chance for improvement.

Please find below my few Specific comments:

1. Number of Keywords used in this paper is too many, it needs shortening

2. Abstract is too long ,please adhere to Journal rule and regulation (world count < 300)

3. Methods : The study design was not clear and who is the provider of medical care for the migrants. Is there any difference between native and migrants in terms of facility they visit and level of professional involved and logistic and supply related issue?, Is migrants have equal access to clinical care as native people?

4. How is the Danish health system function means ,who cover the cost of medical care? is it free or out of pocket payment ?

5. Discussion: There is discrepancy in terms of monitoring, biochemical level and use of diabetic lower agents and medication use in the two groups but why this happen was not clear and is migrants living in another country also have similar finding. Was their education level and income included in the analysis?

Reviewer #4: The manuscript is well written and can be published

Reviewer #5: This is a very interesting paper.

i have a few comments for the authors to address.

1. The design of the study was not stated. Please state the study design and this should reflect in the abstract.

2.Line 48: " A study population for the analysis of T2D was defined by excluding ................. The study population must be explicitly defined by the authors before stating the exclusion criteria.

3. The sampling approach employed in the retrieval of participants' health records from the data base was not clearly stated. Please briefly describe your method of selection (Sampling method). This is very important to clear doubts about the possibility of the authors using data that had the outcomes they wanted).

4. How did you arrive at the study size? What was the size of the sampling frame?

5. How was the prevalence of T2D assessed or calculated.

6. Line 65: "information on hospital admission and outpatient contacts from 1994 onward..........." What does onward here mean? Does it mean from 1994 to 2022???? Please state the specific relevant dates or period of data extraction and please make the necessary changes from line 68 -78.

7. I belief the possibility of medical data entry errors may be slim in advanced countries like Denmark, but that possibility cannot be ruled out. Please discuss how you dealt with this issue and its possibility of confounding the outcomes/results.

Reviewer #6: Comments

The study is interesting as it assesses the disparities in prevalence, monitoring, and implementation of clinical guidelines for type 2 diabetes among host and migrant populations. The paper is well written however these few comments need to be addressed.

Abstract

In a population-wide, register-based T2D population of 254,097 individuals across nine origin groups (Native Danish, Middle East, Europe, Turkey, Former Yugoslavia, Pakistan, Sri Lanka, Somalia, Vietnam) care was assessed in 11 indicators corresponding to CG recommendations.

Comment: The statement is difficult to comprehend, could you kindly revise it to improve readability?

The justification for conducting the study is missing. Authors must include the justification for conducting the study

Line 196: It is not appropriate to commence a statement with numbers. Kindly reword or spell out the number.

Table 1: some of the variables in table 1 are not reported, kindly check.

7. PLOS authors have the option to publish the peer review history of their article (what does this mean?). If published, this will include your full peer review and any attached files.

**Do you want your identity to be public for this peer review?** For information about this choice, including consent withdrawal, please see our Privacy Policy.

Reviewer #2: **Yes: **Jonathan A Abuga (MPH, PhD)

Reviewer #3: **Yes: **Amsalu Bekele Binegdie

Reviewer #4: No

Reviewer #5: No

Reviewer #6: No

---

## [Decision Letter · Decision Letter 2]

9 May 2023

PGPH-D-22-01596R2

Guideline-level monitoring, biomarker levels and pharmacological treatment in migrants and native Danes with type 2 diabetes: population-wide analyses

Dear Dr. Isaksen,

Thank you for submitting your manuscript to PLOS Global Public Health. After careful consideration, we feel that it has merit but does not fully meet PLOS Global Public Health’s publication criteria as it currently stands. Therefore, we invite you to submit a revised version of the manuscript that addresses the points raised during the review process.

The manuscript has been evaluated by two reviewers, and their comments are available below.

The reviewers have raised concerns regarding the reporting and methodology of this study. 

Could you please revise the manuscript to carefully address the concerns raised?

We look forward to receiving your revised manuscript.

Kind regards,

Johannes Stortz

Staff Editor

Journal Requirements:

Additional Editor Comments (if provided):

Reviewers' comments:

Reviewer's Responses to Questions

**Comments to the Author**

1. If the authors have adequately addressed your comments raised in a previous round of review and you feel that this manuscript is now acceptable for publication, you may indicate that here to bypass the “Comments to the Author” section, enter your conflict of interest statement in the “Confidential to Editor” section, and submit your "Accept" recommendation.

Reviewer #7: (No Response)

Reviewer #8: All comments have been addressed

2. Does this manuscript meet PLOS Global Public Health’s publication criteria? Is the manuscript technically sound, and do the data support the conclusions? The manuscript must describe methodologically and ethically rigorous research with conclusions that are appropriately drawn based on the data presented.

Reviewer #7: Yes

Reviewer #8: Yes

3. Has the statistical analysis been performed appropriately and rigorously?

Reviewer #7: No

Reviewer #8: Yes

4. Have the authors made all data underlying the findings in their manuscript fully available (please refer to the Data Availability Statement at the start of the manuscript PDF file)?

Reviewer #7: Yes

Reviewer #8: Yes

5. Is the manuscript presented in an intelligible fashion and written in standard English?

Reviewer #7: Yes

Reviewer #8: Yes

6. Review Comments to the Author

Reviewer #7: See in the attachment

Reviewer #8: The manuscript presents the finding of a register-based study that examines the disparities in T2D care among migrants compared to native Danes in Denmark. The study analyzed 11 indicators of guideline-level care among a large sample including 254,097 individuals with T2D. The migrants were grouped by their countries of origin, and the results showed that in most migrant groups, T2D was more prevalent than in native Danes. The study found that non-fulfillment was common in eight indicators, with monitoring and biomarker control being inferior among migrants compared to native Danes. Migrants from Somalia received the poorest care overall, with elevated risks in all eleven indicators and exceedingly high lipid levels. The study concludes that T2D care is inferior among most migrant groups in Denmark, and there are significant disparities in care, with migrants from Somalia receiving the poorest care. The authors have adequately addressed the reviewer's concerns. My comment is only on the reporting the prevalences. The hospital-based prevalence may not resemble the population-based prevalence. Are there any differences in the proportion of referred patients for the diagnosis of diabetes between native Danes and migrants? The possible reasons of such disparity such as structural racism, genetic predisposition, and lifestyle are the future area of interest.

7. PLOS authors have the option to publish the peer review history of their article (what does this mean?). If published, this will include your full peer review and any attached files.

**Do you want your identity to be public for this peer review?** For information about this choice, including consent withdrawal, please see our Privacy Policy.

Reviewer #7: No

Reviewer #8: No

---

## [Editor Report · Decision Letter 3]

25 Aug 2023

Guideline-level monitoring, biomarker levels and pharmacological treatment in migrants and native Danes with type 2 diabetes: population-wide analyses

PGPH-D-22-01596R3

Dear Mr Isaksen,

We are pleased to inform you that your manuscript 'Guideline-level monitoring, biomarker levels and pharmacological treatment in migrants and native Danes with type 2 diabetes: population-wide analyses' has been provisionally accepted for publication in PLOS Global Public Health.

Best regards,

Julia Robinson

Executive Editor